# Synergy of cations in high entropy oxide lithium ion battery anode

Kai Wang[1,2], Weibo Hua [3], Xiaohui Huang[1,2], David Stenzel[1,2], Junbo Wang[1,2], Ziming Ding[1,2], Yanyan Cui[1,2], Qingsong Wang [1], Helmut Ehrenberg [3], Ben Breitung [1,2], Christian Kübel [1,2,4,5] ✉ & Xiaoke Mu[1] ✉

High entropy oxides (HEOs) with chemically disordered multi-cation structure attract intensive interest as negative electrode materials for battery applications. The outstanding electrochemical performance has been attributed to the high-entropy stabilization and the so-called 'cocktail effect'. However, the configurational entropy of the HEO, which is thermodynamically only metastable at room-temperature, is insufficient to drive the structural reversibility during conversion-type battery reaction, and the 'cocktail effect' has not been explained thus far. This work unveils the multi-cations synergy of the HEO $Mg_{0.2}Co_{0.2}Ni_{0.2}Cu_{0.2}Zn_{0.2}O$ at atomic and nanoscale during electrochemical reaction and explains the 'cocktail effect'. The more electronegative elements form an electrochemically inert 3-dimensional metallic nano-network enabling electron transport. The electrochemical inactive cation stabilizes an oxide nanophase, which is semi-coherent with the metallic phase and accommodates $Li^+$ ions. This self-assembled nanostructure enables stable cycling of micron-sized particles, which bypasses the need for nanoscale premodification required for conventional metal oxides in battery applications. This demonstrates elemental diversity is the key for optimizing multi-cation electrode materials.

Metal oxides anode materials enable conversion reactions and provide high theoretical capacity in lithium ion batteries (LIBs)[1–3]. However, the poor electrical conductivity[4–6] and severe structural disintegration during the reaction[7,8], which result in the notorious 'size effect'[9–11], hinders the materials from practical application except after costly nano-structuring. Design of multi-cation oxides has been demonstrated to be an promising strategy for overcoming the drawbacks in the conversion reaction[12–14]. Adding an additional metal element has been proposed to enhance the electron conductivity of the pristine materials[12,15] and the reaction products[16,17], and also to reduce the volume change while reacting with Li ions[12,16].

Recently, high entropy oxides (HEOs) as a novel class of multi-cation metal oxides have attracted intensive interest for battery applications[18–24]. In particular, it has been reported that micrometer-sized HEO particles show extraordinary long-term cycling stability at high capacity without a necessity for nanostructuring when used as anode material in LIBs[20]. It overcomes the 'size effect' of conventional oxides during the conversion reaction, resulting in a material with promising chances for practical application. The unexpected performance has been attributed to the randomly mixed five metal elements in a single-phase solid solution without chemical short-range order. This has been supposed to result in a high configurational entropy,

[1]Institute of Nanotechnology (INT), Karlsruhe Institute of Technology, Hermann-von-Helmholtz-Platz 1, 76344 Eggenstein-Leopoldshafen, Germany. [2]Department of Materials and Earth Sciences, Technical University Darmstadt, 64287 Darmstadt, Germany. [3]Institute for Applied Materials (IAM), Karlsruhe Institute of Technology, Hermann-von-Helmholtz-Platz 1, 76344 Eggenstein-Leopoldshafen, Germany. [4]Helmholtz-Institute Ulm for Electrochemical Energy Storage (HIU), Karlsruhe Institute of Technology (KIT), Helmholtzstraße 11, 89081 Ulm, Germany. [5]Karlsruhe Nano Micro Facility (KNMF), Karlsruhe Institute of Technology (KIT), Hermann-von-Helmholtz-Platz 1, 76344 Eggenstein-Leopoldshafen, Germany. ✉e-mail: christian.kuebel@kit.edu; xiaoke.mu@kit.edu

stabilizing the crystal structure during lithium storage[18,20,23,24]. However, solely considering the entropy is insufficient to explain the reversibility of the conversion reaction and the high electron conductivity that is required for the long reaction path due to the large particle size. Ghigna et al. recently used operando XAS and realized that the HEO $Mg_{0.2}Co_{0.2}Ni_{0.2}Cu_{0.2}Zn_{0.2}O$ is not reacting fully reversible during electrochemical cycling[25]. This questions the simple high entropy interpretation. The term cocktail effect has been raised by many authors attempting to interpret the observed electrochemical improvements[18,19,26–28]. However, using term cocktail effect does not really bring any understanding for the synergistic effects of the cations during the electrochemical reaction. Why this HEO performs differently from well-studied binary multi-cation oxides is not clear. In this work, we unveiled the synergistic effects of the cations in the HEO $Mg_{0.2}Co_{0.2}Ni_{0.2}Cu_{0.2}Zn_{0.2}O$ during electrochemical reaction with lithium by a detailed analysis of the valance state of the metal elements and a comprehensive characterization of the micro structure of the material at different cycling states of the HEO-based anode using X-ray absorption spectroscopy (XAS) and analytical (scanning) transmission electron microscopy (S/TEM). This work further contributes to the idea of designing multi-cation materials for high performance ion batteries.

## Results

### Valence states and atomic coordination

The HEO micro particles as the active material are prepared as composite electrodes and assembled in coin-type half-cells with a carbonate electrolyte using lithium foil as the counter and reference electrode. The half-cells are cycled in the voltage window of 0.01 V (fully discharged state) to 3.0 V (fully charged state) versus $Li/Li^+$. XAS results including X-ray absorption near edge structure (XANES) and extended X-ray absorption fine structure Fourier transforms (EXAFS-FT) of the K edge of Co, Ni, Cu and Zn at different electrochemical reaction states are shown in Fig. 1. The XANES and EXAFS-FT results show the expected 2+ valence state for Co, Ni, Cu and Zn in the as prepared HEO. The valance state of Mg could not be analyzed, because the energy of the Mg-K edge is below the minimum energy of the hard X-ray spectrum available at the beamline.

After the first discharging, the XANES spectra of Co, Ni, Cu and Zn (solid violet lines in Fig. 1a–d) match well to the corresponding metallic references. This proves that Co, Ni, Cu and Zn have been reduced to the metallic state, fitting to the expected conversion reaction. Consistently, the EXAFS-FTs (solid violet lines in Fig. 1e–h) show that the metal-oxide (M-O) bond distances present in the as-prepared HEO disappeared and metal-metal (M-M) bond distances are present in the discharged sample, suggesting that oxygen was removed from the coordination shell of the Co, Ni, Cu and Zn atoms resulting in a metallic structure. The M-M distances of Co, Ni and Cu agree with metallic *fcc* Co, Ni and Cu. The M-M distances of Zn do not match metallic Zn (*hcp*, space group *P63/mmc*), but fit to Zn-Li distances in *fcc* LiZn. Alloying of Zn and Li during electrochemical cycling is well known and has been reported by various studies of ZnO batteries[29–31].

Interestingly, after recharging to 3.0 V, Ni and Cu stay in the metallic state as determined from the unchanged XANES pre-edges (Fig. 1b, c, red lines) and the M-M distances present in the EXAFS-FTs (Fig. 1f, g, red lines). Consistently, no M-O distances are observed for Ni and Cu in the EXAFS-FTs. The Co XANES reveals a significant reduction of the pre-edge peak (solid red line in Fig. 1a) and the presence of both M-M and M-O distances in the Co EXAFS-FT (Fig. 1e, red line). This indicates that a large fraction of Co participates in the redox reaction while some Co remains in a metallic state. Distinct from the other metals, Zn is almost fully reoxidized to the 2+ state (Fig. 1d, red line). The Zn EXAFS-FT (Fig. 1h, red line) shows that the characteristic Zn-O distances reappear, while the Zn-Zn distances disappear. The atomic structure of the samples was further analyzed by electron pair

distribution function (ePDF) (Supplementary Fig. 5) derived from select area electron diffraction (SAED) patterns. The evolution of the M-O-M and M-O distance and the creation of M-M bonds confirmed the conclusions deduced from the XAS analysis. Details can be found in the Supplementary Fig. 5.

### Micro structure, elemental and phase distribution

S/TEM analysis was applied to understand the micro structure at the atomic level for the different states. The intensity in high-angle annular dark field STEM (HAADF-STEM) images is roughly proportional to the atomic number squared, the atomic density and the sample thickness. For the as-prepared sample (Fig. 2a), HAADF-STEM shows a homogeneous intensity distribution with only a few pores present, whereas the images of the cycled samples (Fig. 2b, c) exhibit a heterogeneous nanostructure. In particular, dendritic features can be clearly observed. This observation indicates a nanoscale phase/elemental separation in the electrode after dis/recharging.

Elemental maps obtained by STEM based electron energy-loss spectroscopy (EELS) mapping explain the HAADF contrast. In the discharged sample (Fig. 2e), Co, Ni and Cu are spatially well-correlated with each other and correspond to the bright areas (dendritic and bright granular features) in the HAADF image (Fig. 2d). O and Mg and to some extent Zn are anti-correlated with Cu, Ni and Co and correspond to the dark regions in HAADF-STEM. For better visual guidance, we overlaid the Cu and O maps in Fig. 2f and highlighted two exemplary regions demonstrating the correlation. The result indicates that the discharged sample consists of a phase containing Cu, Ni and Co in the metallic state and a phase containing O, Mg, and presumably Li (which cannot be clearly distinguished in the EELS signal due to an overlap with the M-edges of the transition metals). The Zn distribution fits neither to the Cu-Ni-Co phase nor the $MgO(Li_xO)$ phase and presumably corresponds to the LiZn distribution, present as indicated by the EXAFS-FT analysis.

The phase separation is mostly maintained in the 1st charged sample (Fig. 2g–i). Different from the discharged case, now Zn exhibits a strong spatial correlation with O and Mg. They form an oxide phase visible as the dark regions in the HAADF image. This is consistent with the 2+ oxidation state observed for Zn in XAS and also suggests that MgO and ZnO are mixed at the sub-nm or atomic level. Cu and Ni are still highly correlated and major features for Co are the same as Cu and Ni. They are anti-correlated with the oxide phase and form a metallic phase as bright granular regions and dendritic features in the HAADF image (Fig. 2c, g). This agrees with Cu, Ni and a large fraction of Co not being oxidized after recharging as determined from the XAS results. However, the granular distribution of Co is less sharp compared to the discharged state, implying some fraction of Co contributing to the oxide.

In the high-resolution TEM (HRTEM) images (Fig. 3) of the cycled sample, the metal phase appears darker compared to the metal oxide phase. For the charged sample, the reflections in the fast Fourier transform (FFT) (Fig. 3c) from a dendritic area (Fig. 3b, blue box) can be indexed as [101] zone axis of a *fcc* metallic structure with space group *Fm-3m*. The FFT (Fig. 3d) of an area inside a grain (Fig. 3b, orange box) consists of two sets of reflections. The one with larger reciprocal space distances can be indexed as [101] zone axis of a metal *fcc* structure with an orientation rotated 59° anti-clockwise to the adjacent dendritic feature at the grain boundary. An enlarged image of the gain boundary is shown in Supplementary Fig. 15. The unit cell parameter measured from the FFTs of the metallic phase (both the dendritic region and in the grain) is 3.6 Å close to *fcc* Cu (3.6 Å), Ni (3.5 Å) and Co (3.6 Å) single elemental metals. It confirms this metallic phase to be a CoNiCu alloy. The other set of reflections in the FFT from withni the grain can be indexed as [101] zone axis of a rock-salt metal oxide. Notably, the metal and the oxide phases in the grain exhibit a semi-epitaxial relationship. The same phenomenon was also observed in the discharged sample (Fig. 3e–g).

Scanning nanobeam electron diffraction (4D-STEM) was carried out to map the structure and phase distribution with a larger field of view compared to HRTEM; details are provided in the experimental section. As an example, Fig. 4 shows a 4D-STEM result of the discharged sample. The crystal orientation and phase map can be obtained by indexing the local diffraction patterns (e.g. Fig. 4d–f). Correlating the HAADF image (Fig. 4a) and the crystal orientation map (Fig. 4b), one can clearly see that the dendritic features, which formed during lithiation, are dominantly located at the grain boundaries (e.g. the ones indicated by the arrows in Fig. 4a).

The nanoscale phase map (Fig. 4c, taken from the location marked by the white dotted box in Fig. 4b) shows the nanoscale separation of the metallic (green) and oxide (red) phase inside the grain and the CoNiCu dendritic-featured phase located at the grain boundaries. The

local diffraction patterns (e.g. Fig. 4d, e, two typical ones taken from the regions marked by the yellow and blue boxes in Fig. 4c) can be indexed to either a simple *fcc* structure with lattice parameters of crystalline Cu or a rock-salt *fcc* structure with lattice parameters of crystalline MgO. Due to the overlap of the phases in projection, diffraction spots from both phases are observed in every diffraction pattern. The colors in the phase map represent the phase dominating the signal. The result is consistent with the STEM-EELS elemental maps and also the XAS, ePDF and HRTEM results. All these indicate the formation of the alloy and oxide phase with lattice parameters very close to Cu and MgO.

In addition of the CuNiCo phase and the oxide phase, a *fcc* LiZn phase is observed in the 4D-STEM data. For example, the diffraction pattern (Fig. 4f) taken from the red box in Fig. 4a with the

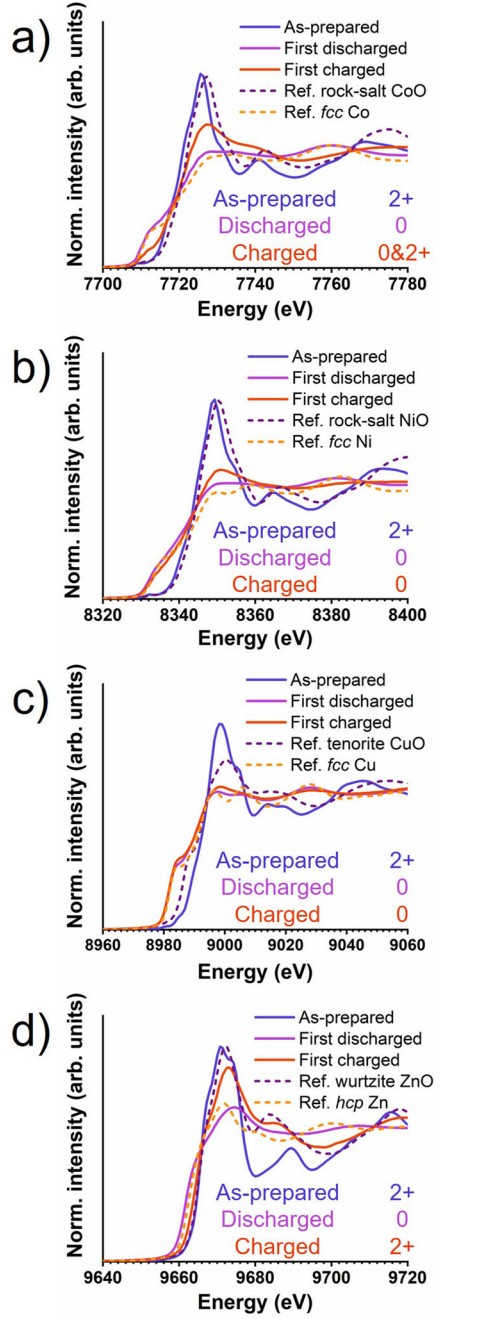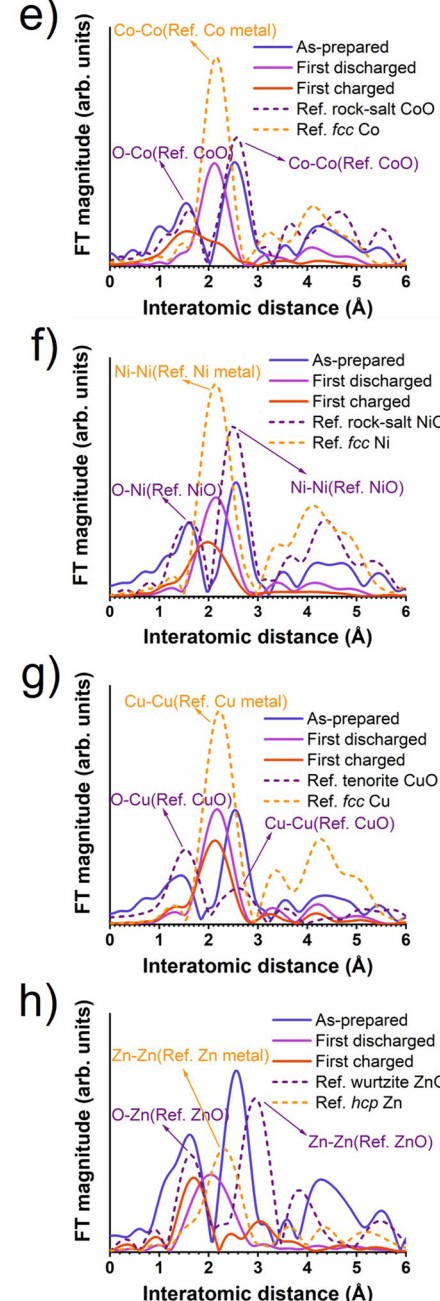

**Fig. 1 | Valence state and atomic coordination analysis. a–d** XANES of Co, Ni, Cu and Zn; **e–h** corresponding EXAFS–FT, solid lines represent the experimental data and dashed lines are references measured from standard samples provided by the beamline. 'First discharged' and 'First charged' sample refers to the samples discharged to 0.01 V and charged to 3.0 V (versus Li/Li⁺).

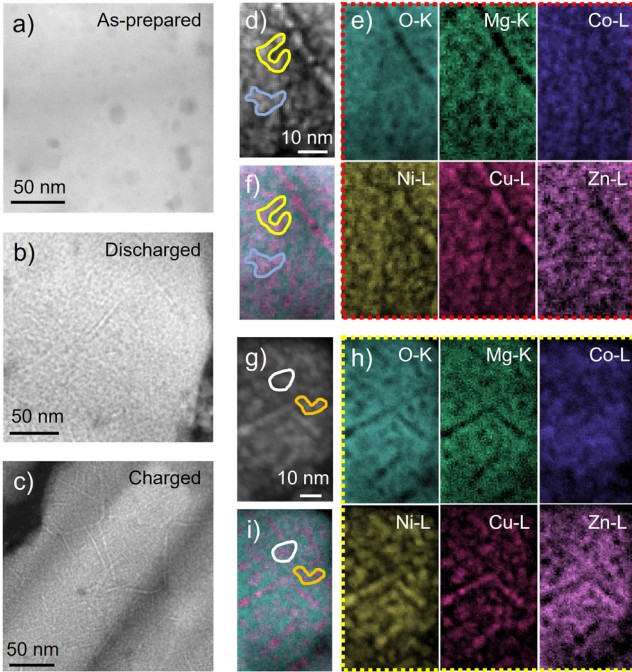

**Fig. 2 | Elemental distribution analysis of the as-prepared and cycled samples.**
**a**–**c** HAADF-STEM images of the as-prepared, 1st discharged (discharged to 0.01 V versus Li/Li$^+$) and 1st charged (charged to 3.0 V versus Li/Li$^+$) samples; **d**, **e** HAADF-STEM image and STEM-EELS elemental maps from an exemplary location of the 1st discharged sample; **g**, **h** HAADF image and EELS elemental maps from the 1st charged sample; **f**, **i** combined O and Cu maps.

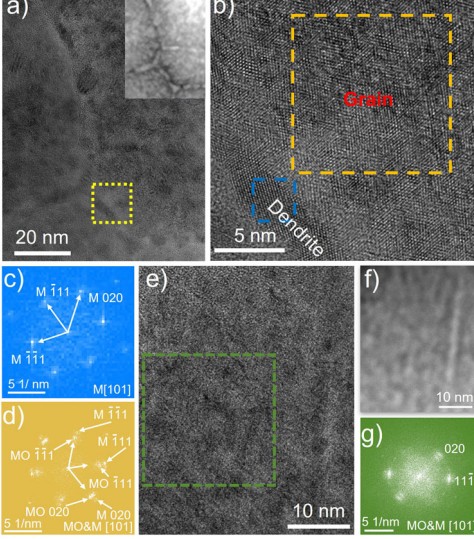

**Fig. 3 | Structural characterization of the cycled samples. a** HRTEM image of the 1st charged sample, and inset showing a HAADF-STEM image from the same area; **b** enlarged image of the area marked in **a**; **c**, **d** FFTs of the areas marked in **b**; **e** HRTEM image of the discharged sample; **f** HAADF-STEM image of the same area as **e**; **g** FFT of the area marked in **e**.

corresponding intensity profiles (Fig. 4g, h) along the arrows show broad diffraction peaks, which can only be explained by overlapping reflections of the *fcc* LiZn phase with the CoNiCu alloy and the oxide phase. While the similarity of the d-values between LiZn and the metal oxide together with the limited resolution in reciprocal space make it difficult to obtain a reliable phase map of LiZn, this manual analysis of

the diffraction patterns (e.g. Fig. 4f) unambiguously reveals the presence of LiZn crystals. Note that, the LiZn exhibits a clearly defined orientation relationship with the CuNiCo and the oxide, giving rise to a single crystalline appearance of the grains in Fig. 4b despite 3 different phases being present.

Due to a large amount of metal atoms reduced during the 1st discharging and only partial reoxidation during the following charging, an equal amount of lithium ions corresponding to the reduction of these metal atoms are expected to exist both in the discharged and the charged material. Although it is difficult to directly detect nano phase Li$_2$O due to its electron beam sensitivity and overlap of the Li-K edge with the M-edge of Cu, Co, Ni and the tail of the volume plasmon, Li$_2$O with a size above a few ten nanometer can be directly observed in TEM, especially using electron diffraction with controlled dose. This has been widely used in many recent studies, even during in-situ TEM experiments[32–34]. Our 4D-STEM and SAED data confirms that no bulk Li$_2$O is present in the cycled material. However, we cannot rule out the presence of Li$_2$O nanophases. Therefore, it is reasonable to deduce that Li$^+$ is incorporated in the oxide phase either as nano/sub-nanometer scale lithium oxide aggregates or atomically dispersed in the rock-salt structure giving rise to a Mg-Li-O configuration in the discharged state and a Mg-Co-Zn-Li-O configuration in the charged state. In a previous study the possibility of atomically dispersed lithium has been demonstrated, where a series of rock salt (Co,Cu,Mg,Ni,Zn)$_{1-x}$Li$_x$O oxides ($x < 0.30$) was successfully synthesized and the ionic conductivity of the materials increases with increasing Li amount[35].

Figure 5d, e illustrates the epitaxial relationship observed for all phases inside the grains using the discharged state as an example. The metallic phase and the oxide phase exhibit the same crystal orientation. The large mismatch of the unit cell parameters (0.36 nm for the CoNiCu alloy and 0.42 nm for the oxide) results in a significant concentration of crystal defects (approximately one dislocation necessary every 7 (200) planes), which can be directly observed in the HRTEM images. The associated structural distortion can explain the disappearance of the crystal-field splitting (at 9670 eV and 9674 eV) corresponding to an octahedral environment in the Zn XANES spectrum of the charged sample (Fig. 1d)[36,37]. The large lattice mismatch finally results in the observed nanoscale phase separation and small crystallite size. The {220} lattices of LiZn (0.22 nm) match the {020} of the oxide phase (0.21 nm), and the {400} lattices of LiZn (0.16 nm) match the {220} of the oxide phase (0.15 nm) as illustrated in Fig. 5d. Zn and partial Co are oxidized to Zn$^{2+}$ and Co$^{2+}$ and merge into the oxide phase in the charged state.

## 3D electron conductive network

Electron tomography was used to visualize the 3-dimensional (3D) distribution of the nanophases. The tomographic reconstruction from the 1st charged HEO is shown in Supplementary Movie 1. Figure 6a shows a visualization from a cropped tomographic reconstruction (the corresponding volume rendering video is shown in Supplementary Movie 2). A series of volume rendered images of the area marked by the white box is shown in Supplementary Fig. 17. The generated video after surface rendering of the grain boundary is shown in supplementary movie 3. Supplementary Movie 4 shows the animation after combining the volume rendering of grain boundary adjacent area and the surface rendering of the grain boundary. From the Supplementary Movie 3 and 4, it can be clearly observed that the cold color part (represents the alloy) has a two-dimension shape. Figure 6b–f is reconstructed slices over a depth range of 51.2 nm normal to the viewing direction in Fig. 6a. The bright parts correspond to the metallic phase due to its significantly higher density than the oxide phase. The dendritic features (e.g. the one indicated by the yellow arrow) are present throughout the depth range revealing their platelet-like shape at the grain boundary instead of a 1-D dendrite structure. The platelets at the boundary of adjacent grains intersect and form a 3D network.

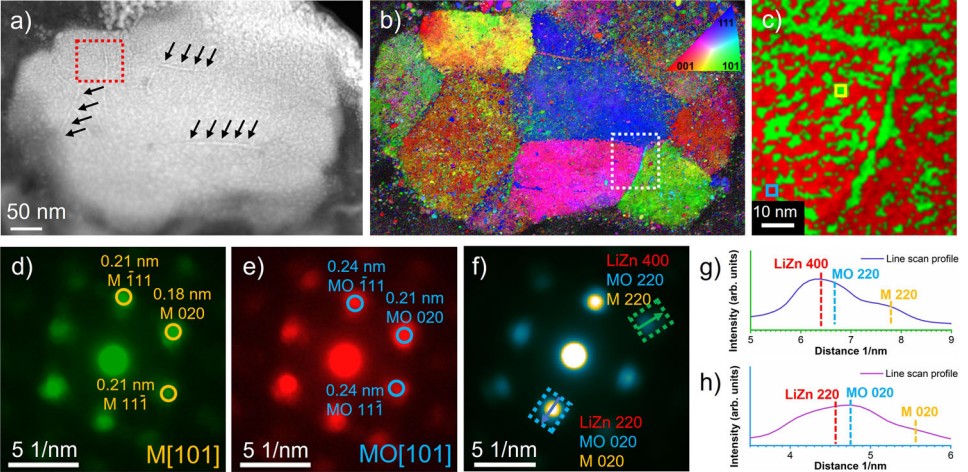

**Fig. 4 | Orientation and phase distribution analysis of the 1st discharged sample.** **a** HAADF-STEM image; **b** orientation map obtained by indexing the diffraction patterns of the 4D-STEM data; **c** typical phase map corresponding to the area marked by the white rectangular box in **b**; **d**, **e** two exemplary diffraction patterns averaged from the area marked by the yellow and blue boxes in **c**, the circles mark the M and MO reflections; **f** diffraction pattern averaged from the area marked by the red dotted box in **a**; **g**, **h** line profiles along the arrow in the green and blue dotted rectangular boxes in **f**.

Together with the metal phase inside the grains, a fine nanoscale 3D conductive network is formed, which penetrates throughout the micrometer-sized primary particle. The oxide phase fills the region between the metal network forming a 3D framework as counterpart.

The metallic network provides a highly efficient electron-transfer path, while the metal oxide containing Li ions provides good ionic conductivity. To confirm this, we measured the conductivity of the as-prepared and the charged microparticles using an in-situ TEM-STM holder following the configuration shown in Fig. 7a. The as-prepared sample exhibits an insulating behavior in the voltage range of −15 to 15 V (Fig. 7b). In contrast, significantly higher currents (0.1 μA) can pass through the charged particle at 15 V, indicating a significantly enhanced electrical conductivity (the high resistance in the voltage range of −5 to 5 V is presumably due to the contact resistance between the tip and particle). We conducted electrochemical impedance spectroscopy (EIS) measurements in a half-cell configuration to separate the contributions of ionic and electrical conductivity and provide more statistics than the microscopic in-situ investigation. One can see a smaller diameter of the semicircle in the EIS spectra of the charged sample (Fig. 7c, red) compared to that in the as-prepared sample (Fig. 7c, black). It confirms the enhanced electrical conductivity in the charged sample.

To confirm that our observations also reflect the structural changes and thus the reaction mechanism during the following cycles, discharged and charged samples were characterized after 5 cycles (Supplementary Figs. 22 and 25). The results reveal the same structural features as seen before for the corresponding states of the 1st cycle. This further confirms the observed high reversibility of the electrochemical reaction after the 1st discharge. Large field of view STEM-EDX maps of the anode in the charged state of the 5th cycle (Supplementary Fig. 5) confirm that no elemental segregation occurs at the micrometer level and that the observed phase separation only occurs at the nanoscale.

## Discussion

### Explanation of the cation synergy

Based on the results shown above, we propose the following microscopic scenario for the reactions. During the 1st discharging, Li ions migrate into the oxide particle. This reduces Co, Ni, Cu and Zn to the metallic state, leading to a phase separation: (i) Co, Ni and Cu form a simple *fcc* structured alloy at the nanoscale, which penetrates throughout the whole micron-sized particle forming a 3D network.

This is expected to contribute a theoretical capacity of 611 mAh/g. (ii) Zn further alloys with Li forming a LiZn nanophase, which provides an additional 76 mAh/g theoretical capacity. The simultaneous reduction and alloying give rise to the single broad redox peak in the CV curve of the 1st discharging process (Supplementary Fig. 26) and a total theoretical capacity of 687 mAh/g. (iii) A Mg-dominated oxide phase with rock-salt *fcc* structure is left as a matrix filling the space in the metal network and accommodating Li⁺ or LiₓO. The LiZn, CoNiCu and MgO(LiₓO) phases exhibit a semi-epitaxial relationship. During charging, Li in LiZn is first delithiated, followed by further oxidation of Zn and oxidation of some of the Co to the 2+ valence state. This theoretically provides about 305 mAh/g capacity (with 50% of the Co participating the charging reaction). The $Zn^{2+}$ and $Co^{2+}$ replace the Li⁺ in the oxide matrix, which migrates into the electrolyte during charging. Afterwards, the delithiation stops. Cu, Ni and some of the Co do not participate in the reaction, maintaining the 3D metallic network in the charged state. This theoretically incurs an irreversible capacity of 382 mAh/g. We noted that the experimental capacity is higher than the theoretical expectation for both the 1st discharging and the following charging processes. This has been widely reported[38,39] for the anode and is typically attributed to the formation and decomposition of the solid electrolyte interphase (SEI)[8,40], pseudo capacitance[41] and a contribution of the conductive carbon[42]. In addition, our characterization cannot rule out the possibility that the CoNiCu phase may also alloy with Li. Nevertheless, the discovery of the irreversible metallic phase explains the observed 400 mAh/g irreversible capacity between the 1st discharging process (~900 mAh/g) and the following charging process (~500 mAh/g).

The initial HEO structure cannot be restored in the recharging process of the 1st cycle. The HEO was synthesized at elevated temperature (>877 °C), where the configuration entropy can effectively stabilize the chemical-disordered crystal structure[43]. It is intuitive that the HEO structure, which is thermodynamically only metastable at room-temperature, cannot be cycled reversibly in the conversion reaction. The outstanding electrochemical performance of the material results from the synergy of the cations, where the five metal elements can be grouped to provide three different functions. (i) Zn and Co are the electrochemically active elements providing the main capacity of the battery from the first lithiation process. (ii) $Mg^{2+}$ is electrochemical inert and cannot be reduced by Li. It stabilizes the oxide phase, which accommodates Li⁺ in the discharged state and hosts $Co^{2+}$ and $Zn^{2+}$ in the charged state. This is consistent with

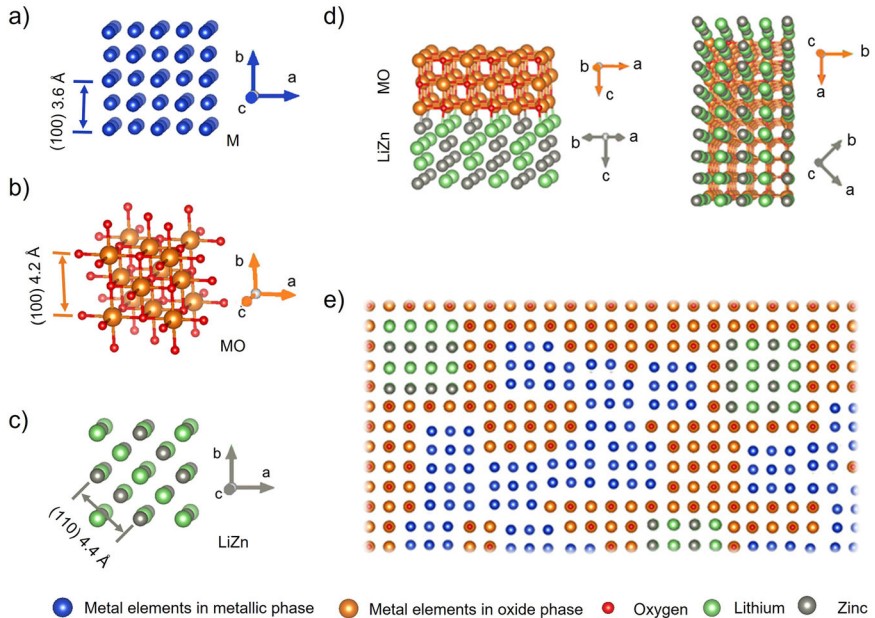

**Fig. 5 | Epitaxial phase relationships in the discharged state. a–c** Sketches of the atomic structure of the metal (M) phase, the metal oxide (MO) phase and LiZn phase; **d** orientation relationship of the oxide phase and LiZn; **e** a schematic overview of the structure.

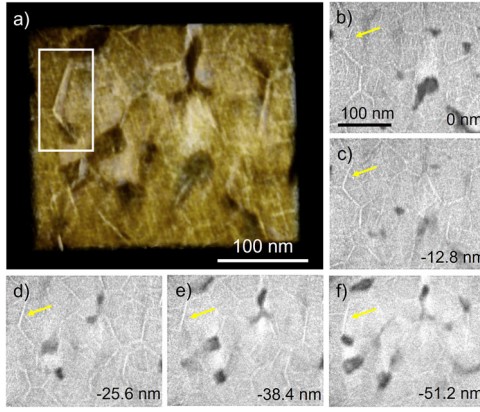

**Fig. 6 | Analysis of the 3D conductive network. a** Volume rendering based on a tomographic reconstruction; **b–f** slices through the reconstructed volume normal to the viewing direction in **a**, the depth is denoted at the bottom right corner in the images.

previous studies, where Qiu et al. proposed the inactive MgO could act to preserve the material preventing pulverization of the electrode[44], which was recently experimentally demonstrated by Wang et al. using the operando synchrotron transmission X-ray microscopy[45]. (iii) Cu and Ni do not participate in the redox reaction after the 1st discharging, but maintain the backbone of the nanoscale 3D metallic network, which provides excellent electron conductivity. This means that a significant fraction of Li+ ions are left in the oxide nanophase after charging to balance for the metals not reoxidized. This ensures good ionic conductivity. Ultimately, the micron-sized material particles turn into a composite with intrinsic semi-coherent metal/oxide nanophases. This provides a reduced reaction path down to the nanometer level. Atoms/ions only need to adjust their position locally along well-defined crystallographic lattices. This not only guarantees a fast reaction, but moreover, decreases the formation of bulk $Li_2O$, which is one of the key reasons for capacity fading in other batteries. The ductile metal backbone presumably also mechanically stabilizes the material and maintains its integrity

despite the volume changes during de/lithiation. All these factors help to protect the micrometer particles from fragmentation during the conversion and alloying reaction, giving the material the behavior of being artificially pre-nanostructured. This ultimately grants the electrode a significantly improved volumetric energy density and reduced cost for industrial production.

Traditional multi-cation oxides consisting of binary metal elements provide either a metallic phase in the charged state for improving the electric conductivity[17,46], or two oxide compounds with different expansion coefficients for buffering the volume change in the conversion reaction[12,14,16]. However, the enhancement is limited, and pre-nanostructuring or a combination with conductive nanomaterials is still required to achieve stable cycling[12]. In contrast, the five-cation oxide possesses cations with diverse distinct properties, giving rise to the metal and oxide phases semi-coherently entangled at the nanoscale with superior benefits over traditional bi-cation oxides. Although it can be clearly seen that the reactions after the 1st discharging are independent of the proposed 'high entropy stabilization', the highly disordered chemical environment of the synthesized HEO is a prerequisite for the entanglement of the nanophases. Our findings can help to design new conversion-type materials for the application as battery electrodes.

## Methods
### Materials synthesis
The HEO ($Mg_{0.2}Co_{0.2}Ni_{0.2}Cu_{0.2}Zn_{0.2}O$) was prepared using nebulized spray pyrolysis (NSP) as reported by Sarkar et al.[20]. The corresponding nitrates (($Co(NO_3)_2 \cdot 6H_2O$ (Sigma Aldrich, 99.9%), $Cu(NO_3)_2 \cdot 2.5H_2O$ (Sigma Aldrich, 99.9%), $Mg(NO_3)_2 \cdot 6H_2O$ (Sigma Aldrich, 99.9%), $Ni(NO_3)_2 \cdot 6H_2O$ (Sigma Aldrich, 99.9%), and $Zn(NO_3)_2 \cdot 6H_2O$ (Alfa Aesar, 99.9%)) were dissolved in deionized water. The solution was sprayed as a mist with a speed of about 40 mL/h and blown with $N_2$ as carrier gas into the hot zone of a tubular furnace, operated at 1150 °C. At the hot-wall reactor (assembled at the end of tubular furnace) at elevated temperature, the precursor solution transforms into the desired crystalline oxide.

### Electrochemical measurements
The HEO was mixed with Super C65 carbon black and polyvinylidene fluoride (PVDF) with a mass ratio of 7: 2: 1 dispersed in N-methyl-2-

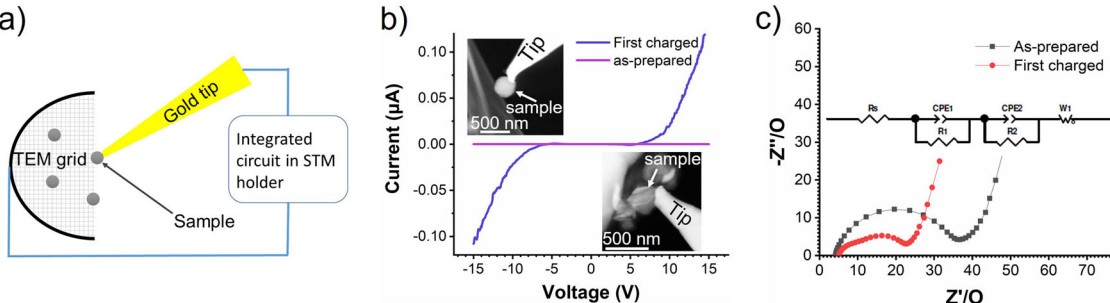

**Fig. 7 | Conductivity test. a** Schematic setup of the in-situ conductivity measurement (a full sketch of the TEM-STM holder is drawn in Supplementary Fig. 19); **b** the current-voltage curves of the as-prepared and 1st charged sample, the insets at the top-left and bottom-right show STEM images corresponding to the measurement of the as-prepared and the charged sample; **c** electrochemical impedance spectroscopy (EIS) results of the as-prepared and charged sample, the equivalent circuit is shown in the inset.

pyrrolidone (NMP). A high-energy dispersion machine (ARE-250, Thinky Corporation, 3 min, 2000rpm) was used to make a uniform slurry. The resulting slurry was pasted on a copper film current collector with a thickness of 8 μm and the electrode dried in a vacuum oven at 80 °C overnight. The HEO mass loading of the electrodes for TEM and XAS investigation was about 0.6 and 10 mg/cm², respectively. The electrode material was cut into discs with a diameter of 13 mm to be used as the working electrodes. The coin-type half-cell was assembled in a glove box (water and oxygen partial pressure below 0.5 ppm) using glass microfiber filter paper (GF/C, Whatman) with a diameter of 17 mm and Li metal foil (Gelon LIB Co., Ltd) (diameter 13 mm, thickness 1 mm) as separator and counter electrode. The electrolyte was 1 M/L lithium hexafluorophosphate (LiPF6) in a 3: 7 weight mixture of ethylene carbonate/ethyl methyl carbonate (Selectilyte LP57, BASF SE). The amount of the electrolyte used in each half-cell was about 120 μL. Galvanostatic measurements were performed using a battery test system (BT-2000, Arbin Instruments) at 25 °C. The half-cell was first discharged (lithiation) to 0.01 V, and then charged (deliatiation) to 3.0 V using a specific current of 50 mA/g on the basis of HEO mass. The half-cell cycling was stopped at the targeted state and the material dissembled from the electrodes and prepared for XAS and TEM investigation. Electrochemical impedance spectroscopy measurements (EIS) were performed using a VSP-300 (BioLogic Science Instruments) potentiostatic device. The EIS data was acquired by applying 10 mV AC perturbation signal over a frequency range of 1 MHz to 1 Hz (6 points per decade). The simulation was conducted using the Zview software (Ametek Scientific Instruments). The equivalent circuit used for the simulation of the 1st charged sample is shown in Fig. 7c. Rs can be attributed to the real part of the impedance arising from the electrolyte resistances. The first semicircle in the high frequency region is caused by the SEI layer formation in the cycled material, which introduces a charge transfer resistance (R1) and an ion diffusion resistance (constant phase element, CPE1) at the SEI/electrolyte interface. R2 and CPE2 represent the electron and ion resistance at the anode/SEI interface. Wo is the Warburg impedance, related to chemical diffusion processes. For the simulation of the as-prepared sample, the equivalent circuit of the 1st charged sample was used, but the first resistant (R1) and constant phase element (CPE1) parallel circuit were removed.

The theoretical capacity was calculated using the equation:

$$Q = \frac{nF}{M_w}$$

where $Q$ is the theoretical capacity, $n$ is the number of transferred charges per formula unit during the electrochemical reaction, $F$ represents the Faraday constant (96485 C/mol), and $M_w$ is the molecular weight of one formula unit.

## X-ray absorption spectroscopy

X-ray absorption spectroscopy (XAS) measurements were performed at the XAS beamline of the synchrotron radiation source KARA at the Karlsruhe Institute of Technology (KIT), beamline P64 and P65 at PETRA III, Germany. The XAS data were recorded in both fluorescence and transmission mode. X-ray absorption near edge spectra (XANES) of the XAS spectrum were obtained by subtracting the pre-edge background from the overall absorption and normalizing to the spline fit using the ATHENA software package. The k2-weighted extended X-ray absorption fine structure (EXAFS) was Fourier transformed over the limited k-range from 3 to 10 Å⁻¹ with a hanning window ($dK = 1$ Å⁻¹).

## Transmission electron microscopy (TEM)

The necessary TEM lamellae were prepared by focus ion beam (FIB) lift-out using Strata 400 S (ThermoFisher Scientific) at 30 kV, and further polished at 2 kV to remove potential surface damage caused by the high energy Ga ions. The TEM images, SAED and EELS measurements in Supplementary Fig. 14 were acquired using an aberration (image) corrected Titan 80–300 (FEI Company) microscope operated at 300 kV, equipped with an UltraScan (Gatan Inc.) CCD camera and Tridium 863 (Gatan Inc.) imaging filter (GIF). The EELS mapping in Fig. 2 and Supplementary Figs. 12, 13 and 22, EDX mapping in Supplementary Fig. 24, the 4D-STEM and the HRTEM measurements were performed using a double (probe and image) corrected Themis Z (ThermoFisher Scientific) microscope operated at 300 kV, equipped with Oneview CMOS camera (Gatan Inc.) and a continuum imaging filter with K3 IS camera (Gatan Inc). The EELS maps were acquired with a 0.7 nm scan step, 30 mrad convergence angle, 35 mrad collection angle and 250 pA probe current using the K3 IS direct electron camera and continuum camera. The EELS maps shown in Fig. 2 and Supplementary Fig. 22 were denoised by principal component analysis (PCA) using the multivariate statistical analysis (MSA) plug-in in DigitalMicrograph (Gatan Inc.). Caution was payed to avoid over-fitting the data by keeping the significant higher order components, in which only noise can be seen, in the reconstruction.

Electron tomography was performed using a 2020 tomography holder (Fischione Instruments) on a probe-corrected Themis 300 (ThermoFisher Scientific) microscope operated at 300 kV in STEM mode. HAADF-STEM tilt series were recorded over a range of ±72° with a tilt step of 2°. Projections were aligned in IMOD (http://bio3d. colorado.edu/imod/index.html) using 6.5 nm Au colloidal particles as fiducial markers with a mean residual alignment error of 0.45 pixels. The aligned tilt series were reconstructed using the simultaneous iterative reconstruction technique (SIRT) within Inspect3D (ThermoFisher Scientific). 3D visualization was performed in Avizo 2020.2 (ThermoFisher Scientific).

Electron pair distribution function (ePDF) results were derived from the SAED patterns obtained using parallel beam condition.

Multiple frames were integrated to increase the intensity on the diffraction ring at approx. $0.21\,nm^{-1}$ (corresponding to the {002} lattice of the rock-salt metal oxide and the {111} lattice of the *fcc* metal) to more than 30,000 $e^-$ per pixel to provide sufficient single to noise ratio. The diffraction pattern was azimuthally integrated to yield a 1D diffraction profile, denoted as $I(s)$, $s = \theta/\lambda$, where $\theta$ is the scattering angle and $\lambda$ is the wavelength of the high energy TEM electrons. Following the procedure in literature[47–49], the diffraction profile was normalized to $\varphi(s) = \frac{I(s) - N\langle f(s)^2 \rangle}{N\langle f(s)\rangle^2} s$, with $\langle f(s)\rangle$ the averaged atomic scattering factor, $N$ the number of atoms involved in electron diffraction, which was determined by minimizing $\langle f(s)\rangle$ at the maximum of $s$ ($s_{max}$). The ePDF is obtained by Fourier sine transformation of $\varphi(s)$, through $ePDF(r) = \int_0^{s_{max}} \varphi(s)\sin(2\pi sr)ds$.

The 4D-STEM mapping was performed using a quasi-parallel beam in STEM mode with a convergence semi-angle of 0.85 mrad and a camera length of 580 mm. These settings result in a probe size of approximately 2 nm. The electron probe was laterally scanned over the region of interest with a step size of 0.8 nm and diffraction patterns were recorded on the OneView (Gatan Inc.) camera (frame size binned to 256 × 256 pixels) at each scan position with an exposure time of 5 ms per diffraction pattern. Crystal orientation and phase maps were obtained by indexing the diffraction patterns of the 4D-STEM dataset using the Automated Crystal Orientation Mapping software (Nano-MEGAS SPRL) [50].

The conductivity of individual particles was measured using a double tilt in-situ TEM-STM holder equipped with a PicoFemto electrical measurement system (Zeptools Technology Co., Ltd.). The detailed testing procedure can be found in the caption of the Supplementary Fig. 19.

## Data availability

The source data used in this study are available at the KITOpen database under access code 1000154295 [https://doi.org/10.5445/IR/1000154295].

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

## Acknowledgements

K.W. (201706240159), X.H. (201806340078), J.W. (201806950082) and Y.C. (201809370066) acknowledge financial support by the Chinese Scholarship Council (CSC), W.H. acknowledges the National Science Foundation of China (grant no. 22108218). This work contributes to the research performed at the Center for Electrochemical Energy Storage Ulm-Karlsruhe (CELEST) and the Helmholtz Joint Laboratoy Model Drive Materials Characterization (MDMC). The authors thank the KNMFi (Karlsruhe Nano Micro Facility) and the POLIS Cluster of Excellence "Post Lithium Storage" (EXC 2154 – Project number 390874152). X.M. thanks the Deutsche Forschungsgemeinschaft for funding of grant MU 4276/1-1. B.B. acknowledges support from EnABLES and EPISTORE funded by the European Union Horizon 2020 research and innovation program under Grant Agreement Nos. 730957 and 101017709.

## Author contributions

C.K. and K.W. initialized the project. X.M. designed and supervised the experiments and data analysis. D.S., J.W. and B.B. provided the materi-als. K.W. prepared the TEM samples and performed the TEM measure-ments, X.H. performed the 3D tomography and Z.D. did the in-situ electron conductivity tests. W.H. and H.E. contributed the XAS analysis. K.W., Y.C. and Q.W. performed the EIS electrochemical performance testing and contributed to the data analysis. X.M., K.W. and C.K. carried out the main interpretation of the results. K.W. and X.M. wrote the first draft of the manuscript. All authors have contributed to the discussion and revision of the manuscript.

## Funding

## Competing interests

The authors declare no competing interests.
