## [Peer Review File · Nature Communications]

Reviewer comments, first round –

Reviewer #1 (Remarks to the Author):

I was asked to review the manuscript "Synergy of cations in high entropy oxide lithium ion battery anode" written by K. Wang et al.

In this paper, the Authors investigate the reaction mechanisms of the prototypical high entropy oxide as an anode for Li-ion batteries, by employing XAS and advanced microscopic techniques. The work is well presented and leads to interesting new insights into the lithiation/delithiation mechanism in this material. In my opinion, this manuscript is suitable for publication after some critical points are solved:

- My first comment deals with the novelty of this work. The mechanism of this reaction has been already investigated in literature in several papers. In particular, operando XAS and operando TEM microscopy have been already applied in this system. It was demonstrated that the conversion reaction is not completely reversible, leading to a mixture of metals/metal oxides, and that the residual capacity is due to alloying/dealloying between Li/Zn. Almost all the papers agree that Mg is not involved in the reaction but mostly acts as a matrix, and somehow limits the volume expansion and severe cracking which often affect anodes of this kind. From these previous works, it was evident that the different cations have different roles in the mechanism, and that their combination is beneficial for the electrochemical performances with respect to the binary oxides. However, the Authors add an additional piece to this puzzle, allowing to clarify and visualize, through advanced microscopic techniques, the complex network which is formed upon cycling. In particular, they can provide an explanation on how all these phases, namely the metals, the LiZn alloy and the MgO-based matrix, are closely intertwined and correlated. I recognize the validity of this work, but I strongly suggest the Authors to consider the previous work on this topic, by opportunely comparing and discussing their results in view of what is already present in literature. In this sense, I invite them to consider, along with the papers already cited in the bibliography, also:

- P. Ghigna et al., "Lithiation mechanism in high-entropy oxides as anode materials for Li-ion batteries: an operando XAS study", ACS Appl. Mater. Interfaces (2020), 12, 50344-50354

- Wang S-Y. et al., "Operando synchrotron transmission X-ray microscopy study on (Mg, Co, Ni, Cu, Zn)O high-entropy oxide anodes for lithium-ion batteries", Mater. Chem. Phys. (2021), 274, 125105

And I also suggest a freshly published review that sums up most of the research progress for HEO as anodes.

- X. Liu et al., "High-entropy oxide: A future anode contender for lithium-ion battery", EcoMat. 2022, e12261

- All the microscopic investigation is carried out with S/TEM, with a resolution of hundreds of nanometres at the maximum. Do the Authors have some proof that their results are valid in all the regions of the material? It is possible that the electrochemical cycling leads to inhomogeneity and segregation of some phases. The Authors should provide a SEM image coupled to EDS after discharging and/or charging to prove that the material is homogeneous and, if not, provide some statistics and identify the regions where their results are valid.

- Fig. 1 d shows the XANES spectra at the Zn K-edge. Not only the EXAFS, but also the XANES region is indicative of the coordination geometry around the photoabsorber. It can be observed that, before cycling, the spectrum shows the splitting of the white line typical of Zn²⁺ in an octahedral environment, as expected for the pristine single-phase rock salt structure. After the first cycle, however, the splitting is no more observed, and the overall shape seems that of ZnO in the wurtzite structure. From the discussion, it seems that Zn²⁺ is inserted in the MgO matrix, but in that case a rock-salt environment would be expected. A clarification about this aspect should be added in the paper.

Also, this directly connects with the previous remark: Zn can be oxidized to ZnO as wurtzite, leading to segregation in some region of the sample. It is mandatory to gain information on a larger scale to have a complete picture of this system.

- In this system, due to the large degree of irreversibility, the first cycle is drastically different from the following ones. In order to unequivocally assess the mechanism of lithiation/delithiation, two cycles at least should be investigated. Are the results still valid after the first cycle? The Authors should provide some information, if not for all the second cycle, at least for the second lithiation process. I understand that a replication of all the measurements for the second cycle is not feasible in a reasonable time, but at least some microscopic investigation (e.g. HAADF/STEM) should be added to prove that the 3D network (semi-epitaxial dendrite + oxide matrix) is preserved in the following cycles.

- Can the Authors rule out that part of Mg forms an alloy with Li during lithiation, similarly to Zn?

- Line 105, "we overlaid the Cu and Mg maps in 106 Figure 2f", but the caption says that Cu and O maps were overlapped. Please correct.

Reviewer #2 (Remarks to the Author):

Regarding the considered paper I, in fact, have mixed feelings. On one side, it presents comprehensive evaluation of the attractive and novel candidate high entropy anode material for Li-ion cells, with experiments being well-performed, and discussion based on the results of properly selected, advanced experimental methods. But at the same time, the conclusions provided by the Authors are, in my opinion, not fully supported by the obtained results. In particular:

1) No data are provided regarding characterization of the samples beyond the first lithiation and the following delithiation. In my opinion, despite known from literature good cycling ability of the $\text{Mg}_{0.2}\text{Co}_{0.2}\text{Ni}_{0.2}\text{Cu}_{0.2}\text{Zn}_{0.2}\text{O}$ HEO anode (which, however, depends on the synthesis method and preparation of the anode layer), this is not sufficient to be sure that the described by the Authors microscopic scenario of the electrochemical reactions is really correct regarding stable cycling performance. If this should be limited to the first cycle, it is not sufficient, in my opinion, to warrant publication in the Nature Communications journal.

2) Discussion about the anticipated and measured capacity seems simplified too much. There is a large irreversible capacity observed between the first lithiation, which corresponds i.a. to the decomposition of the initial structure, and the following delithiation. Please provide detailed calculations regarding the capacity for the lithiation and delithiation. What is the actual reversible capacity for the used current density? Also, as much as 20 wt.% of carbon black was also used to prepare the electrodes, which contribution to the capacity should be also taken into account.

3) Voltage differences used in the measurements concerning resistance are beyond the voltage range of the electrochemical reaction. Can the sample remain (locally) stable under such experimental conditions?

4) In the discussion section, the Authors wrote that formation of the bulk Li_2O is prevented for the considered anode. But in my opinion, such statement cannot be so simply justified based on the reported data, even for the first cycle only. In fact, it is not described well what happens with the lithium responsible for the irreversible capacity, and as Li_2O is generally poorly observable in many techniques, presence of Li_2O cannot be, in my view, excluded.

Overall, I suggest major corrections.

To reviewer 1:

I was asked to review the manuscript “Synergy of cations in high entropy oxide lithium ion battery anode” written by K. Wang et al. In this paper, the Authors investigate the reaction mechanisms of the prototypical high entropy oxide as an anode for Li-ion batteries, by employing XAS and advanced microscopic techniques. The work is well presented and leads to interesting new insights into the lithiation/delithiation mechanism in this material. In my opinion, this manuscript is suitable for publication after some critical points are solved:

1. My first comment deals with the novelty of this work. The mechanism of this reaction has been already investigated in literature in several papers. In particular, operando XAS and operando TEM microscopy have been already applied in this system. It was demonstrated that the conversion reaction is not completely reversible, leading to a mixture of metals/metal oxides, and that the residual capacity is due to alloying/dealloying between Li/Zn. Almost all the papers agree that Mg is not involved in the reaction but mostly acts as a matrix, and somehow limits the volume expansion and severe cracking which often affect anodes of this kind. From these previous works, it was evident that the different cations have different roles in the mechanism, and that their combination is beneficial for the electrochemical performances with respect to the binary oxides. However, the Authors add an additional piece to this puzzle, allowing to clarify and visualize, through advanced microscopic techniques, the complex network which is formed upon cycling. In particular, they can provide an explanation on how all these phases, namely the metals, the LiZn alloy and the MgO-based matrix, are closely intertwined and correlated. I recognize the validity of this work, but I strongly suggest the Authors to consider the previous work on this topic, by opportunely comparing and discussing their results in view of what is already present in literature. In this sense, I invite them to consider, along with the papers already cited in the bibliography, also:

- P. Ghigna et al., “Lithiation mechanism in high-entropy oxides as anode materials for Li-ion batteries: an operando XAS study”, ACS Appl. Mater. Interfaces (2020), 12, 50344-50354

- Wang S-Y. et al., “Operando synchrotron transmission X-ray microscopy study on (Mg, Co, Ni,Cu, Zn)O high-entropy oxide anodes for lithium-ion batteries”, *Mater. Chem. Phys.* (2021), 274, 125105

And I also suggest a freshly published review that sums up most of the research progress for HEO as anodes.

- X. Liu et al., “High-entropy oxide: A future anode contender for lithium-ion battery”, *EcoMat.* 2022, e12261

We thank the reviewer very much for the recognition of the advance of our work and the suggested relevant publications to improve the discussion of the manuscript. As is partially evident from the publications mentioned above, there has been a significant interest to unravel the reaction mechanism of the HEO in LIBs. While we recognize the importance of the previous work, we do believe that our work is providing a real leap in understanding the reaction mechanism. The reported new results fully revealed the synergy of the cations in the electrochemical reaction, which has only been discussed vaguely as simple “cocktail effect” before. It has now been unraveled microscopically at the nanoscale and atomic level, identifying the different contributions to stability, electrical/ionic conductivity and electrochemical capacity by the different cations.

We appreciate the reviewer’s kind suggestion and have included the above mentioned publications in the revised manuscript. More discussions introducing their findings have been added in the introduction (page 3 line 41 for the review paper from X. Liu et al. and lines 49-51 for the work from P. Ghigna et al). We also compare our observations with these works when discussing our results. In pages 13-14, lines 291-294, we cited Wang S-Y. et al paper which directly supports our findings of the synergy for the role of Mg.

2. All the microscopic investigation is carried out with S/TEM, with a resolution of hundreds of nanometres at the maximum. Do the Authors have some proof that their results are valid in all the regions of the material? It is possible that the electrochemical cycling leads to inhomogeneity and segregation of some phases. The Authors should provide a SEM image coupled to EDS after discharging and/or charging to prove that the material is

homogeneous and, if not, provide some statistics and identify the regions where their results are valid.

We performed EELS analysis (as discussed in the manuscript) at several different areas for each sample at different (de)lithiation states. All the results are consistent. In addition, all STEM images acquired from the FIB and ultramicrotome prepared samples (1st discharged and 1st charged samples) show the same features for different particles reflecting the statistical validity of the phase segregation as discussed in the manuscript. To confirm and visualize this and address the reviewer's concern, we performed large-field-of-view STEM-EDX analysis after the 5th cycle (charged HEO). We added the new results as Supplementary Figure 24 (also attached below for your convenience). Figure S24a is a STEM-HAADF image showing the field of view. The maps (figure S24b) reveal a uniform distribution of all elements in each particle, indicating that there is no phase separation at the micrometer level, only the nanoscale phase separation discussed in detail in the paper and also revealed by the STEM-HAADF image in Figure S24c taken at higher magnification from the location indicated by the dashed box in figure S24a.

Supplementary Figure 24. Large area STEM-EDX map of HEO after 5 cycles in the charged state prepared by FIB: **a)** STEM-HAADF image, **b)** EDX maps and **c)** enlarged image of the area marked in **a)**. No phase separation can be detected at this length scale. The pixel size of the map is 18.6 nm and field of view is 4.4 μm \times 2.8 μm .

The STEM images acquired at randomly selected HEO particles in the 1st charged state are shown below (figure R1), which exhibit the dendritic features characteristic for the nanoscale phase separation. The observation of the nanoscale phase separation is statistically significant and dominates all analyzed particles.

Following reviewer's comment, we have added the new results to the main manuscript (page 12, lines 255-257).

Figure R1: STEM-HAADF images of randomly selected HEO particles in the 1st charged state.

3. Fig. 1 d shows the XANES spectra at the Zn K-edge. Not only the EXAFS, but also the XANES region is indicative of the coordination geometry around the photo absorber. It can be observed that, before cycling, the spectrum shows the splitting of the white line typical of Zn²⁺ in an octahedral environment, as expected for the pristine single-phase rock salt structure. After the first cycle, however, the splitting is no more observed, and the overall shape seems that of ZnO in the wurtzite structure. From the discussion, it seems that Zn²⁺ is inserted in the MgO matrix, but in that case a rock-salt environment would be expected. A clarification about this aspect should be added in the paper.

Also, this directly connects with the previous remark: Zn can be oxidized to ZnO as wurtzite, leading to segregation in some region of the sample. It is mandatory to gain information on a larger scale to have a complete picture of this system.

The authors thank the reviewer for the careful evaluation of the data. Following this comment, we carefully re-examined the 4D-STEM maps and selected area electron diffraction (SAED) patterns taken from different places of cycled samples. No diffraction dots corresponding to the ZnO Wurtzite structure have been found with both manual analysis and template-based indexing methods (Nanomegas). The new EDX maps with large field of view also show no Zn segregation in the cycled samples at the micron level. Considering that all Zn ions participate in the reaction, one can conclude that the majority of Zn ions are not in the Wurtzite structure.

We agree that the XANES reflects the local atomic environment, and the splitting white line is a characteristic feature of an octahedral coordination. The fact that this splitting was not observed in the recharged sample is indicating that the environment is not a perfect rock salt structure. This does not conflict with our other results. As we see in the diffraction patterns and HRTEM images, the recharged sample has a highly defective structure with significantly reduced long range order. With Zn incorporated in the oxide network, a strong lattice distortion is expected. The lattice distortion affects the crystal-field splitting (CFS) as pointed out by previous studies (F. Groot et al., Phys. Rev. B 40, 5715–5723 (1989); F. Kutzler et al., Phys. Rev. B 29, 6890–6900 (1984)). In the case of the cycled material, the local distortion of the Zn-O coordination in a very random manner results in various local environments of Zn and further leads to a broadening of the XANES peaks without clear CFS. This presumably results in the splitting not being observed in the XANES of the recharged sample. Following the reviewer's suggestion, we added the explanation in the manuscript on pages 9-10 *lines 203-206*.

4. In this system, due to the large degree of irreversibility, the first cycle is drastically different from the following ones. In order to unequivocally assess the mechanism of lithiation/delithiation, two cycles at least should be investigated. Are the results still valid after the first cycle? The Authors should provide some information, if not for all the second

cycle, at least for the second lithiation process. I understand that a replication of all the measurements for the second cycle is not feasible in a reasonable time, but at least some microscopic investigation (e.g. HAADF/STEM) should be added to prove that the 3D network (semi-epitaxial dendrite + oxide matrix) is preserved in the following cycles.

We followed the reviewer's comment and carried out new STEM-HAADF, STEM-EELS and HRTEM measurements on the material after 5 cycles. Figures below (Supplementary Figure 22a and 25a) show STEM-HAADF images at the 5th charged and the 5th discharged state revealing the nanoscale phase separation analogous to the 1st charged and discharged state. Supplementary Figure 22b and c show the EELS maps acquired from the 5th charged sample. It shows that (i) O, Mg, Zn are spatially correlated indicating an Mg/Zn oxide phase; (ii) Ni and Cu are correlated forming a metallic phase. (iii) Co has a more homogeneous distribution than other elements indicating it joins both the oxide and metallic phases. HRTEM images and their FFTs (Supplementary Figure 22d, 22e, 23 and 25b and 25c) show that the epitaxial relationship between the oxide and metallic phase is maintained at both the discharged and charged states after 5 cycles. The results agree well with the structure after the 1st cycle and demonstrate that the 3D network is preserved throughout the following cycles. We have updated the main text to include these new results to strengthen this work (page 12, lines 251-255). As additional supplementary information, the voltage-capacity profiles during the 1st to the 5th electrochemical cycles are also shown below (Supplementary Figure 21).

Supplementary Figure 22. (S)TEM, elemental distribution and crystal structure analysis of a sample after 5 cycles showing very similar features as seen previously after the first cycle: **a)** HAADF-STEM image of the sample after the 5th recharging; **b)** EELS elemental maps; **c)** combined O and Cu maps; **d)** HRTEM (enlarged section from Supplementary Figure 23); **e)** FFT of the image shown in Supplementary Figure 23.

Supplementary Figure 25. (S)TEM analysis of a 5th cycled sample in the discharged state showing very similar features as seen previously after the first discharge: **a)** STEM-HAADF image, **b)** HRTEM image and **c)** corresponding FFT.

Supplementary Figure 21. The voltage-capacity profiles during the 1st to 5th electrochemical cycling reveal a good reversibility after the 1st cycle. The difference between the 1st discharge capacity and the discharge capacity in the following cycles can mainly be attributed to the irreversible oxide reduction and SEI formation.⁵ From the 1st recharging to 5th recharging, the capacity do not change significantly indicating reversible electrochemical processes in following battery cycles.

5. Can the Authors rule out that part of Mg forms an alloy with Li during lithiation, similarly to Zn?

We noticed that an average voltage of lithiated phase Li_xMg was reported at 32.5 mV against Li in M. N. Obrovac and V. L. Chevrier, *Chem. Rev.* 2014, 114, 23, 11444–11502. But this requires the present of metal Mg in the anode. Previous works show that MgO is electrochemically inactive (S. Wang *et al.*, *J. Electrochem. Soc.* 168, 020514 (2021) and J. Chen, *Rare Metals* 30, 166–169) and acts as a passivation layer for Li_xMg -formation (L-L Kong *et al.*, *Adv. Funct. Mater.* 2019, 29, 1808756). Therefore, it is not expected that MgO can transform to Li_xMg during the reaction.

In our experiments, we observe no indication for the formation of a bcc-type Li_xMg phase nor for a hcp-type Mg phase, neither in the diffraction patterns nor in HRTEM images. Furthermore, the EELS maps show that Mg has the same distribution as O, so we have

no indication for the reduction of Mg and formation of Li_xMg in our experiments. With this we can confirm that the vast majority of the MgO stays in the oxide state.

6. Line 105, “we overlaid the Cu and Mg maps in 106 Figure 2f”, but the caption says that Cu and O maps were overlapped. Please correct.

We corrected the typo in the manuscript. It is the overlapping of O and Cu maps.

To reviewer 2

Regarding the considered paper I, in fact, have mixed feelings. On one side, it presents comprehensive evaluation of the attractive and novel candidate high entropy anode material for Li-ion cells, with experiments being well-performed, and discussion based on the results of properly selected, advanced experimental methods. But at the same time, the conclusions provided by the Authors are, in my opinion, not fully supported by the obtained results. In particular:

1) No data are provided regarding characterization of the samples beyond the first lithiation and the following delithiation. In my opinion, despite known from literature good cycling ability of the $\text{Mg}_{0.2}\text{Co}_{0.2}\text{Ni}_{0.2}\text{Cu}_{0.2}\text{Zn}_{0.2}\text{O}$ HEO anode (which, however, depends on the synthesis method and preparation of the anode layer), this is not sufficient to be sure that the described by the Authors microscopic scenario of the electrochemical reactions is really correct regarding stable cycling performance. If this should be limited to the first cycle, it is not sufficient, in my opinion, to warrant publication in the Nature Communications journal.

Thanks for your comment. We fully understand your concern about our observations reflecting the stable cyclic performance with only the experiments after the first cycle. Reviewer #1 has raised the same concerns and we have performed additional measurements after the 5th cycle for clarification. The new results clearly show that the

phenomena observed for the 1st cycle are maintained in the following cycles. Details are included and discussed above in the response to reviewer #1 question #4.

2) Discussion about the anticipated and measured capacity seems simplified too much. There is a large irreversible capacity observed between the first lithiation, which corresponds i.a. to the decomposition of the initial structure, and the following delithiation. Please provide detailed calculations regarding the capacity for the lithiation and delithiation. What is the actual reversible capacity for the used current density? Also, as much as 20 wt.% of carbon black was also used to prepare the electrodes, which contribution to the capacity should be also taken into account.

We thank the reviewer for providing the constructive suggestion. This promoted us to review our discovery (the irreversible metallic phase) and realize that it explains the experimentally observed irreversible capacity between the 1st discharging process and the following cycles to a large extent. Following your suggestion, we provide an estimate of the capacities for each process: for the 1st discharging process, the reduction of Co, Ni, Cu and Zn theoretically contributes 611 mA h/g. The ZnLi alloying process contributes further 76 mA h/g. Totally, the electrochemical reaction of the material is expected to provide about 687 mA h/g theoretical capacity. The remaining capacity observed experimentally in the 1st discharging should be due to the formation of the solid electrolyte interphase (SEI) and possible pseudo-capacitance due to the large surface of the carbon additives. For the 1st charging process, the electrochemical reaction due to the dealloying of ZnLi and oxidation of Zn and Co (with an estimation of 50% Co participates in the charging reaction) can contribute about 305 mA h/g theoretically. Therefore, the theoretical irreversible capacity due to the formation of the metallic phase is 382 mA h/g, which is in good agreement with the experimentally observed irreversible capacity between the 1st discharge and the 1st charge process (about 400 mA h/g as shown in Supplementary Figure 21). The experimental reversible capacity is about 500 mA h/g in the 1st charging process, and quickly reduced and stabilized at 460 mA h/g from the 3rd cycle. It is about 155 mA h/g higher than the estimated theoretical capacity. This can be attributed to the decomposition of the SEI, a possible pseudo-capacitance and a contribution of carbon black used in this work as pointed by the reviewer. According to

previous research (e.g. L. Fransson et al., Journal of Power Sources 101 (2001) 1-9 and Hu, J. et al., Journal of Power Sources 508, 230342 (2021)), carbon black itself has a reversible capacity of about 180 mAh/g. Taking into the weight ratio of 7 (active material) :2 (carbon black) used in this work, the carbon black could contribute a capacity of about 51 mA h/g.

We added the detailed capacity explanation in the revised manuscript in page 13 lines 263-282. We appreciate the reviewer's suggestion for strengthening the work!

3) Voltage differences used in the measurements concerning resistance are beyond the voltage range of the electrochemical reaction. Can the sample remain (locally) stable under such experimental conditions?

First, we would like to clarify that the samples for the ex-situ S/TEM and XAS measurements (which are the main results of this work) are cycled in the voltage window of 0.01 to 3.0 V. The key conclusions have been drawn based on those measurements.

We are aware that the voltage applied for the in-situ measurements is much higher than that used for battery cycling. This is caused by the contact resistance between the tip and the particle. This is a common challenge for this kind of test setup and has been reported in previous publications (e.g. Hao Z. et al., Nano Lett. 14, 4245–4249 (2014); Yang Z. et al., Adv. Energy Mater. 6, 1600806 (2016); Liu, X. et al., Nat. Mater. 20, 1485–1490 (2021); Zhong L. et al., Nano Lett. 13, 2209–2214 (2013)). This has been explained in our manuscript. Estimating the actual voltage across the HEO particle is very difficult. Therefore, to confirm the result from the in-situ STEM experiment and to obtain bulk information, we further carried out the EIS measurements using the as-prepared and 1st charged sample (Figure 6h). The conductivity changes observed both in in-situ testing using the STM holder and in EIS are consistent. In addition, STEM imaging did not indicate any obvious changes during the reaction time of the in-situ measurements.

4) In the discussion section, the Authors wrote that formation of the bulk Li₂O is prevented for the considered anode. But in my opinion, such statement cannot be so simply justified based on the reported data, even for the first cycle only. In fact, it is not described well what happens with the lithium responsible for the irreversible capacity, and as Li₂O is generally poorly observable in many techniques, presence of Li₂O cannot be, in my view, excluded.

Although it is difficult to directly observe nano phase Li₂O due to its electron beam sensitivity and the overlap of the Li-K edge and the Cu, Co, Ni M-edge and the volume plasmon tail, Li₂O with particle sizes above a few ten nanometers can be directly observed in TEM, especially using electron diffraction. This has been widely used for many recent studies, even for the more complex situation during in-situ TEM battery measurements (for example Li, J. et al., Nat Commun 10, 2224 (2019); Zhang, Y. et al., Nano Lett. 14, 7161–7170 (2014); Yang, Z. et al., Small 14, 1803108 (2018)). Our 4D-STEM and SAED data confirmed that there is no formation of bulk Li₂O in the cycled samples. Despite that, we cannot rule out the formation of Li₂O nanophases. We thank the reviewer for pointing out the lack of clear description of the Li⁺ incorporation. We revised our manuscript (pages 8-9 lines 176-190 and page 14 lines 296-297) to better clarify this.

Reviewer comments, second round –

Reviewer #1 (Remarks to the Author):

The authors have addressed all my queries, providing the additional data that, in my opinion, were necessary to fully support their conclusions on the lithiation mechanism of this high entropy oxide. Therefore, I recommend publication of this manuscript.

Reviewer #2 (Remarks to the Author):

The Authors have corrected the submission taking into consideration all given comments, so in my opinion the paper can be accepted as is.